# Evaluating effectiveness of self-help groups in reduction of stigma in patients with neglected tropical diseases in Southern Nigeria: A cluster randomised study

Chinwe C. Eze[1]*, Ngozi Ekeke[1], Wim van Brakel[2], Edmund Ndudi Ossai[3], Precious Henry[1], Amaka Onyima-Esmai[4], Charles C. Nwafor[1], Ngozi Murphy-Okpala[1], Anthony Meka[1], Okechukwu Ezeakile[1], Martin Njoku[1], Francis S. Iyama[1], Chukwuma Anyaike[5], Daniel Egbule[1], Joseph Chukwu[6]

1 Programs Department, RedAid Nigeria, Enugu, Nigeria, 2 Technical Department, NLR-Until No Leprosy Remains, Amsterdam, Netherlands, 3 Department of Community Medicine, Alex-Ekwueme Federal University Ndufu-Alike, Abakaliki, Ebonyi State, Nigeria, 4 Tuberculosis, Leprosy and Buruli Ulcer Control Programme, Ogbaru Local Government Area, Atani, Anambra State, Nigeria, 5 Department of Public Health, Federal Ministry of Health, Abuja, Nigeria, 6 West African Region, German Leprosy and TB Relief Association, Enugu, Nigeria

* chinwendu5000@gmail.com, chinwe.eze@redaidnigeria.org (CCE)

## Abstract

### Introduction

Neglected tropical diseases (NTDs), notably leprosy and Buruli ulcer (BU), often lead to visible impairments and disabilities. Consequently, individuals affected by these conditions face stigmatization, discrimination, and mental health disorders. Stigma is particularly prevalent in leprosy, affecting self-esteem, social participation, self-efficacy, and overall quality of life. Various interventions have been developed to mitigate leprosy-related stigma, including individual and group counseling, peer support, economic empowerment initiatives, and establishment of self-help groups. However, rigorous evaluation of the various interventions through randomized controlled trials is lacking, especially within the Nigerian context, and none has included persons affected by BU. To address this gap, effectiveness of combined leprosy and BU self-help groups (SHGs) on stigma reduction was assessed as part of a larger randomized controlled study.

### Methodology/principal findings

A cluster randomized study was conducted in Southern Nigeria from 2021 to 2023. Five local government areas (LGAs) were selected for each of the intervention and control arms. People affected by leprosy or BU were organized into SHGs in the intervention LGAs. Control LGAs were maintained on the existing standard of care, which did not include SHGs. Monthly peer-support meetings were held for 24 months

**Data availability statement:** The data that support the study's findings are available on figshare at the following link: https://figshare.com/articles/dataset/ LRI_data_Baseline_Repository_xlsx/28302872.

**Funding:** The project was funded by the Leprosy Research Initiative (LRI). The funders had no role in study design, data collection and analysis, decision to publish, or preparation of the manuscript.

**Competing interests:** The authors have declared that no competing interests exist.

following standardized guidelines after appropriate training of SHG members. The intervention targeted perceived and experienced stigma. Pre- and post-intervention stigma assessments were conducted using the stigma assessment and reduction of impact (SARI) scale in both intervention and control LGAs.

A total of 635 persons affected by leprosy or BU were recruited—362 in the intervention group and 273 in the control group. At baseline, respondents in the intervention group had a significantly higher adjusted mean total score on the SARI Stigma Scale compared to the control group, with an adjusted mean difference of 10.75 (95% CI: 8.05–13.46, $p < 0.000$). However, post-intervention, the adjusted mean SARI Stigma Score significantly decreased in the intervention group compared to the control group, with an adjusted mean difference of 37.72 (95% CI: 36.01–39.43, $p < 0.000$).

## Conclusion/significance

Peer-support and psychosocial support through SHGs significantly contributed to stigma reduction among persons affected by Leprosy/BU. SHGs can play a key role in stigma reduction interventions within NTD programs, contributing to the global leprosy target of achieving zero stigma and discrimination.

Trial Registration: ISRCTN Registry: ISRCTN 83649248. https://trialsearch.who.int/ Trial2.aspx? TrialID = ISRCTN83649248.

## Introduction

Over a billion people worldwide, the majority of whom are living in developing countries, are affected by neglected tropical diseases (NTDs) [1]. These diseases are associated with poverty and have devastating health, social, and economic consequences [1]. In Nigeria, over 122 million people are at risk of NTDs and the country is endemic for most of the NTDs including leprosy and Buruli Ulcer (BU) [2].

NTDs, notably leprosy and Buruli ulcer, often lead to visible impairments and disabilities [3]. Consequently, individuals affected by these conditions face stigmatization, discrimination, and mental health disorders. Major drivers of stigma among persons affected by NTDs include the infectious nature of most NTDs, visible disabilities and the fear of contagion [4]. NTD-related stigma can be enacted (explicit acts of prejudice against persons affected by NTDs) or internalized or self-stigma (a deep sense of inferiority and overwhelming fear of enacted stigma by persons affected by NTDs) [5]

Self-stigma is particularly prevalent in leprosy, affecting self-esteem, social participation, self-efficacy, mental well-being, health-seeking behavior and outcome, and overall quality of life [6–7]. Similarly, studies have confirmed a high level of self-stigma among persons affected by BU [5,8]. Nonetheless, persons affected by leprosy and BU also experience other important forms of stigma, including disclosure concerns, anticipated stigma, and enacted stigma.

In line with the WHO definition of health as not just the absence of disease but a "state of complete physical, mental, and social well-being", addressing the social and

psychological implications of NTDs is necessary for the optimum health of affected individuals. Additionally, addressing stigma and discrimination will contribute to the acceleration of the achievement of the zero leprosy targets [9] and ultimately the achievement of NTD elimination targets, as they can undermine NTD control efforts.

Several interventions have been developed and tried to mitigate leprosy-related stigma. These interventions include individual and group counseling, use of testimonies, participatory videos, comics made by people affected by leprosy in a contact event, peer support, economic empowerment initiatives, self-help groups, and awareness campaigns focusing on human rights [10–13]. However, only one study has systematically evaluated the effect of three (counseling, contact event, and socioeconomic development) of these interventions in a randomized controlled trial [6,12]. In addition, none of these interventions included individuals affected by BU.

This paper focuses on self-help groups (SHGs), defined as "voluntary small group structures for mutual aid in the accomplishment of a specific purpose...usually formed by peers who have come together for mutual assistance in satisfying a common need, overcoming a common handicap or life-disrupting problem, and bringing about desired social and/or personal change" [14] SHGs have been recognized as an integral part of chronic disease management [15–16]. They offer informational and psychosocial support, reduce social isolation, and connect patients and caregivers to others with similar health issues which may lead to greater improvements in overall well-being, symptom control, and coping skills [16–17]. A study conducted in Nepal among leprosy patients suggested a positive relationship between participation in SHGs and reduced stigma [18]. However, due to its cross-sectional design, conclusive evidence of causality could not be established.

Our study aimed to evaluate the effect of combined leprosy and Buruli ulcer self-help groups in reducing stigma among persons affected by these conditions. The study is part of a larger randomized controlled study in Southern Nigeria on the impact of a community-centered approach on improving the mental health and quality of life of individuals affected by leprosy or BU.

## Method

### Setting

Nigeria is made up of six geo-political zones with 36 states and a Federal Capital Territory, which are subdivided into 774 Local Government Areas (LGAs). The National Tuberculosis, Leprosy, and Buruli Ulcer Control Program (NTBLCP) is responsible for the leprosy and BU program in Nigeria. The basic management unit for leprosy and BU data is the LGA, and a local government TB, leprosy, and BU supervisor (LGTBLS) oversees leprosy-related activities within each LGA.

The study was conducted in 10 LGAs, across six states in south-east (Anambra and Ebonyi) and south-south (Akwa Ibom, Bayelsa, Delta, and Cross River) geo-political zones. The five LGAs with the highest notification figures for leprosy five years prior the start of the intervention were Ebonyi, Etinan, Ogoja, Ethiope East, and Obubra. Incidentally, these LGAs also host facilities that served as former leprosy settlements. Similarly, the five LGAs with the highest notification figures for BU disease were Ogbaru, Anambra East, Ogbia, Isoko South, and Calabar South. The activities were carried out in health facilities and communities where persons affected by Leprosy or BU reside.

### Description of intervention

Persons affected by leprosy or BU were organized into self-help groups (SHGs). The number of SHGs per LGA was determined by the number of patients in the LGA, with some LGAs having more than 2 SHGs. Each group had an average of 10 members. Membership was voluntary. The purpose of forming the SHGs was to reduce stigma and improve their mental well-being and quality of life through peer support. Each group had a leader and a secretary, selected by the group members from among themselves. The leader presides over meetings, while the secretary takes notes and submits reports afterward.

During the formation of SHGs, orientation on the objectives and procedures for the establishment of SHGs was provided by the research team. Through an interactive discussion, feedback from persons affected by leprosy/BU informed essential revisions on the protocol and expected deliverables of the SHGs in line with project objectives. Thereafter, the standard operating procedures (SOP) for the SHGs were finalized. Members of the SHGs were guided to elect/appoint their group leaders for effective coordination.

Members of the SHGs were enlightened on the causes of leprosy and BU, common myths and misconceptions were discussed and clarified. They were trained to provide psychosocial support to one another through:

1. **Early Recognition and Referral**: Identifying group members with signs suggestive of depression or anxiety—such as sudden withdrawal from social gatherings, meeting absences, or sharing experiences possibly related to mental illness during group meetings—and ensuring their referral to trained lay counselors, either directly or through SHG leaders, for further action. Lay counselors included respected members of the community such as religious leaders, youth leaders, women leaders and teachers

2. **Monthly Meetings**: Participating in monthly meetings to share personal challenges and strategies for overcoming them. They also share successes and improvements made since the previous meeting.

3. **Regular Interaction**: Facilitating regular interaction between SHG leaders/members and healthcare workers or community lay counselors for mental health assessment and provision of psychosocial support if necessary.

4. **Feedback and Engagement**: Securing the interest of members through regular feedback from SHG leaders to the research team regarding patient satisfaction with mental healthcare received from lay counsellors and health workers.

Additionally, members organized home visits to those who were inconsistent in attending meetings, participated in skills acquisition activities and saving schemes.

A total of 33 SHGs were established in the intervention LGAs. The intervention lasted for two years from July 2021 to June 2023. Data was collected at baseline and post intervention.

The control LGAs were maintained on the existing standard of care, which did not include SHGs.

## Study design

The study was designed and reported in accordance with the CONSORT 2010 guidelines (see S1 checklist and S1 Fig). A cluster-randomized trial design was employed for the study. Using data from routine surveillance records of the NTBLCP, ten LGAs with the highest number of notified leprosy or BU cases between 2014 and 2018 were selected consecutively from 220 LGAs in southeast/south-south Nigeria.

The study team used a computer-generated list of random numbers to allocate LGAs to intervention and control groups. Each LGA served as a cluster and the unit of randomization. Five clusters (LGAs) namely, Ebonyi, Ogoja, Ogbaru, Ogbia, and Calabar South were randomly selected as intervention groups and the other five LGAs – Etinan, Ethiope East, Obubra, Anambra East and Isoko South served as the control group (Fig 1). After randomization, the intervention LGAs were not contiguous with the control LGAs.

## Study population

The study population comprised persons affected by leprosy or BU who had been registered for treatment in the ten selected LGAs.

**Sample size and sample size calculation.** The study was conducted as part of a larger intervention study focusing on the impact of a community-centered approach to improving the mental health and quality of life of individuals affected by leprosy or Buruli ulcer (BU) in southern Nigeria.

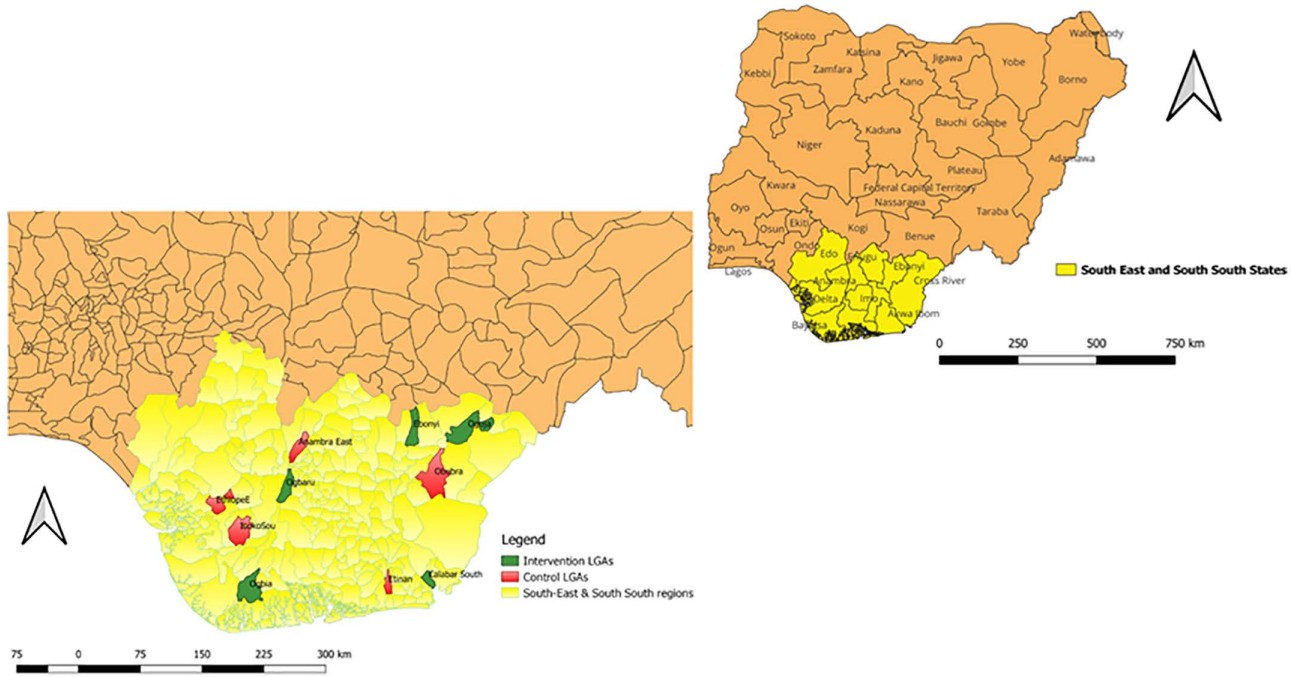

**Fig 1. Map showing the intervention and control LGAs.**

Based on a prevalence rate of 58% for mental health disorders among persons affected by leprosy found in Nigeria [19], the study anticipated a pre-intervention prevalence of poor mental health of 58%, with an expected reduction to 38% post-intervention. This reduction target was informed by a study in India which demonstrated that the use of trained community lay workers for mental health interventions could reduce prevalence of depression by more than 20% [20].

Based on a sample size calculation for 80% power, a significance level of 0.05, and an anticipated 30% loss to follow-up, a minimum sample size of 400 patients (200 in each group: intervention and control) were needed.

It is worth noting that the calculation was based on leprosy data, as information on mental health among Buruli ulcer patients was unavailable.

### Recruitment of participants

To recruit study participants, names and contact details of leprosy or BU patients were extracted from LGA Leprosy and BU Treatment Registers, respectively. Eligible persons were contacted either by phone calls or home visits through the LGA Tuberculosis and Leprosy Supervisors (LGTBLS) and invited for an orientation meeting on a scheduled date.

All leprosy/BU patients who were eligible were enrolled in the self-help groups (SHGs) from the start of the intervention until one year before its end. Recruitment of participants started 1st July 2021 and ended 30th June 2022. Patients who came after the deadline were allowed to join the SHGs and receive appropriate services as stated in the protocol [21], but were not included in the final research analysis.

**Inclusion criteria.** Any person affected by leprosy registered for treatment including children who are old enough to participate and share their experiences.

Any person affected by Buruli ulcer disease including children who are old enough to participate and share their experiences.

All persons affected by leprosy or BU registered for treatment up to one year before the end of the study

**Exclusion criteria.** Refusal to give consent.

Patients who needed urgent medical attention.

Patients unable to communicate clearly.

## Intervention outcomes

The primary outcome of the intervention was reduction in stigma among persons affected by leprosy or Buruli ulcer (BU). The four key domains of stigma were targeted namely- Internalized stigma, Anticipated stigma, Experienced stigma and Disclosure concerns. Changes in stigma were evaluated by comparing mean domain scores and total stigma scores between intervention and control groups at baseline and endline, using both within-group and between-group analyses.

## Data collection

A paper-based data collection method was employed for this study. Trained research assistants administered the Stigma Assessment and Reduction of Impact (SARI) scale to participants both before and after the intervention to gauge their stigma levels. The SARI Stigma Scale is a validated tool originally designed to measure perceived and experienced stigma among leprosy patients. However, research has shown that it is adaptable and applicable to other stigmatizing conditions, such as BU. This tool is best suited for our study as it measures multiple dimensions of stigma, offering a more comprehensive assessment than tools that focus only on self-stigma. It has also been previously used and validated in low-resource settings with populations similar to our study context, demonstrating good psychometric properties. For example, in a study conducted in Indonesia, the SARI Stigma Scale demonstrated good psychometric properties, with a Cronbach's alpha of 0.88 for the total scale and intraclass correlation coefficient of 0.75, indicating strong internal consistency and reliability [22].

The scale consists of 21 items grouped into subdomains that capture key dimensions of stigma, including internalized stigma, perceived discrimination, disclosure concerns, and social exclusion. Each item is scored on a Likert scale from 0 for strongly disagree to 3 for strongly agree. The total score ranges from 0 to 63, with higher scores indicating greater levels of stigma.

Socio-demographic data of participants were also recorded. Data collection was conducted across both intervention and control groups. Although the study had a good retention rate, some of the participants in both groups assessed at baseline relocated or died.

While the study was primarily quantitative and focused on analysis of SARI scale scores, informal feedback and shared experiences from participants and Self-Help Group (SHG) leaders emerged organically during regular SHG meetings. These were not collected through structured qualitative interviews or focus group discussions and therefore are not reported as formal qualitative data. However, relevant insights are referenced in the discussion to provide contextual understanding of participant experiences.

## Data analysis

Data entry and analysis were done using International Business Machine, Statistical Product and Service Solutions (IBM-SPSS) software version 25. Descriptive statistics such as means and percentages were calculated to describe the population. An unpaired, cross-sectional analysis of the baseline and post-intervention data was conducted. Categorical variables were compared between the study and control groups using the Chi-square test, while the means of continuous variables were compared using the Student t-test. Within-group comparisons of the outcome variables used the Student t-test, or a Mann Whitney U test when the data was skewed. To investigate the effect of the intervention on stigma, adjusted mean scores of the SARI scale overall and separately for the four SARI scale domains (internalised stigma, experienced stigma, anticipated stigma, and disclosure concerns) were calculated using mixed-effects models, including random intercepts for clusters. The random intercept in mixed model analysis assesses the likelihood that the clusters

vary significantly from each other in terms of their overall effect on the outcome. A redundant intercept variance indicates that the clusters have similar average outcome, hence whatever difference is observed is likely due to individual level variance and/or intervention effect, rather than differences between clusters. This approach allows for more accurate, unbiased, and generalizable estimates of the intervention's effect by accounting for baseline differences and within-cluster correlations, leading to more reliable conclusions about the intervention's impact. Adjusted mean differences between the intervention and control groups were calculated pre and post intervention. A backward elimination method was used to investigate the factors influencing the effect of the intervention in a multivariate regression model. P-values less than.05 were considered significant.

## Ethical approval

Ethical approval was obtained from the Health Research and Ethics Committee of the University of Nigeria Teaching Hospital, Ituku Ozalla, Enugu (NHREC 0501-2008B). Consent was also obtained from the respective states' Tuberculosis, Leprosy and Buruli ulcer Program Managers and Local Government Tuberculosis, Leprosy and Buruli ulcer supervisors.

Prior to participation, respondents either signed or thumb printed a written informed consent form, and the extent of their involvement in the study was clearly communicated to them. For respondents under 18 years of age, written informed consent was obtained from their parents or guardians, accompanied by verbal assent from the respondents, which was witnessed by the parent or guardian.

Participation in the study was voluntary, and respondents were assured that refusal to participate or withdrawal from the study after giving consent would not result in any form of victimization. Furthermore, respondents were guaranteed that all information provided through the questionnaire would be kept confidential.

## Results

**Socio-demographic characteristics of the respondents.** Table 1 presents the socio-demographic characteristics of respondents in both the intervention and control groups, assessed both before and after the intervention. The intervention group had 362 respondents at baseline, while the control group had 273. Post-intervention, due to attrition, the number of respondents decreased to 286 and 257 respectively in the intervention group and control group. At baseline, significant differences were observed between the intervention and control groups in most assessed socio-demographic variables, except for gender. However, post-intervention, significant differences were found only in age, disease entity and vocational training between the groups. While the mean age at baseline did not differ significantly between the groups, post-intervention analysis revealed a significantly higher mean age in the control group (42.9 years, 95% CI: 40.8–45.0) compared to the intervention group (p = 0.001).

## Impact of the intervention on stigma

Post-intervention, participants in the intervention group experienced a significant reduction in stigma scores across all domains and in the overall mean stigma score. In contrast, the control group showed a significant increase in stigma levels, further highlighting the effectiveness of the intervention.

Fig 2 and Fig 3 compared the levels of stigma between the study and control groups, while Fig 4 provided a within-group comparison, at baseline and post-intervention, across the 4 domains of the SARI stigma scale and the total SARI stigma score.

A comparison of baseline and post-intervention scores on outcome variables for each group is presented in Table 2, while results of the multivariate regression analysis for baseline and post-intervention are presented in S1 Table and S2 Table

**Experienced stigma.** The score for experienced stigma decreased significantly post-intervention in the intervention group with a mean difference of 9.8 (95% CI 8.8-10.6) (Table 2). Conversely, the control group showed a statistically significant increase in experienced stigma score post-intervention compared to baseline at a mean difference of 8.6 (95%

**Table 1. Socio-demographic characteristics of the study and control groups at baseline and post-intervention.**

| Variable | Baseline | | | Post-intervention | | |
|---|---|---|---|---|---|---|
| | Study group (n = 362) N (%) | Control group (n = 273) N (%) | p-value | Study group (n = 286) N (%) | Control group (n = 257) N (%) | p-value |
| **Age of respondents** | | | | | | |
| Mean±(SD) | 44.3±18.3 | 43.2±15.0 | 0.418* | 42.9±18.2 | 47.8±16.6 | 0.001* |
| 95%CI | 42.4-46.1 | 41.1-45.0 | | 40.8-45.0 | 45.7-49.8 | |
| **Age of respondents in groups** | | | | | | |
| <18 years | 33 (9.1) | 13 (4.8) | <0.001 | 29 (10.1) | 10 (3.9) | 0.003 |
| 18-39 years | 110 (30.4) | 93 (34.1) | | 87 (30.4) | 61 (23.7) | |
| 40-49 years | 65 (18.0) | 76 (27.9) | | 62 (21.7) | 74 (28.8) | |
| 50-59 years | 60 (16.6) | 52 (19.0) | | 49 (17.1) | 39 (15.2) | |
| ≥60 years | 94 (26.0) | 39 (14.3) | | 59 (20.6) | 73 (28.4) | |
| **Gender** | | | | | | |
| Male | 181 (50.0) | 141 (51.6) | 0.681 | 135 (47.2) | 123 (47.9) | 0.878 |
| Female | 181 (50.0) | 132 (48.4) | | 151 (52.8) | 134 (52.1) | |
| **Marital status** | | | | | | |
| Never married | 106 (29.3) | 90 (33.0) | < 0.001 | 87 (30.4) | 57 (22.2) | 0.058 |
| Married | 214 (59.1) | 127 (46.5) | | 142 (49.7) | 134 (52.1) | |
| Separated/Divorced | 13 (3.6) | 31 (11.4) | | 15 (5.2) | 25 (9.7) | |
| Widowed | 29 (8.0) | 25 (9.2) | | 42 (14.7) | 41 (16.0) | |
| **Religion** | | | | | | |
| Christianity | 338 (93.4) | 269 (98.5) | 0.003 | 282 (98.6) | 251 (97.7) | 0.533 |
| Islam | 1 (0.3) | 0 (0.0) | | 1 (0.3) | 3 (1.2) | |
| Traditional religion | 23 (6.4) | 4 (1.5) | | 3 (1.0) | 3 (1.2) | |
| **Educational attainment of respondents** | | | | | | |
| No formal education | 124 (34.3) | 55 (20.1) | <0.001 | 63 (22.0) | 63 (24.5) | 0.172 |
| Primary education | 130 (35.9) | 138 (50.5) | | 115 (40.2) | 116 (45.1) | |
| Secondary education | 90 (24.9) | 70 (25.6) | | 93 (32.5) | 72 (28.0) | |
| Tertiary education | 18 (5.0) | 10 (3.7) | | 15 (5.2) | 6 (2.3) | |
| **Employment status of respondent** | | | | | | |
| Unemployed | 191 (52.8) | 51 (18.7) | <0.001 | 125 (43.7) | 99 (38.5) | 0.091 |
| Self-employed | 153 (42.3) | 216 (79.1) | | 154 (53.8) | 143 (55.6) | |
| Salaried employment | 18 (5.0) | 6 (2.2) | | 7 (2.4) | 15 (5.8) | |
| **Have had vocational training** | | | | | | |
| Yes | 17 (4.7) | 40 (14.7) | <0.001 | 105 (36.7) | 46 (17.9) | <0.001 |
| No | 345 (95.3) | 233 (85.3) | | 181 (63.3) | 211 (82.1) | |
| **Disease entity** | | | | | | |
| Leprosy | 251 (69.3) | 250 (91.6) | <0.001 | 171 (59.8) | 221 (86.0) | <0.001 |
| Buruli ulcer | 111 (30.7) | 23 (8.4) | | 115 (40.2) | 36 (14.0) | |

P values are based on chi-square ($\chi^2$) test.

* t-test

CI 9.7- 7.5) (Table 2). Additionally, at baseline, the adjusted mean internal stigma score for the intervention group was 2.2 points higher compared to the control group (95% CI: 1.03 to 3.38, p < 0.000) (Fig 2, S1 Table). While post-intervention, the adjusted mean total score for experienced stigma was significantly lower in the intervention group compared to the control group with adjusted mean difference of 15.44 (95%CI 16.21 to 14.67, p < 0.000) (Fig 3, S2 Table).

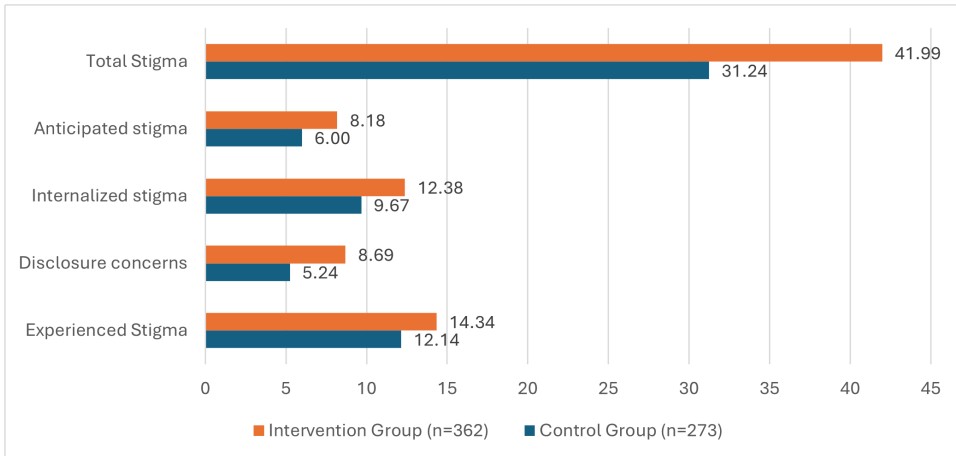

**Fig 2. Adjusted Mean SARI stigma scores at baseline for intervention and control groups.**

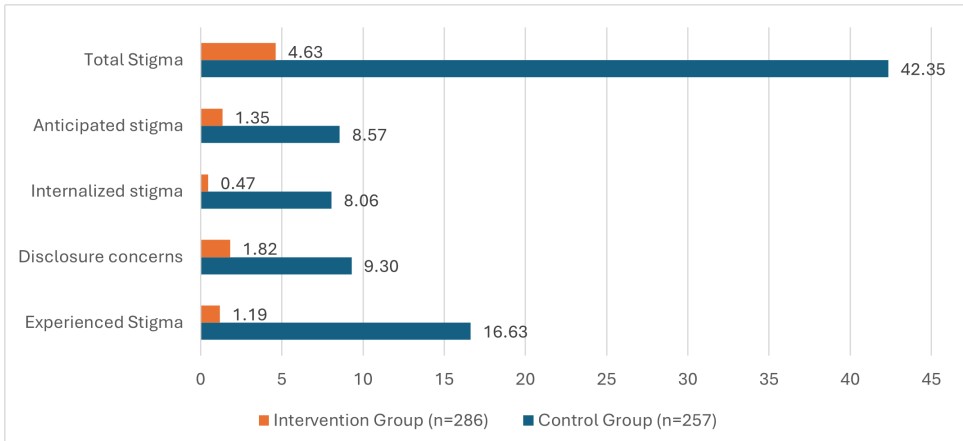

**Fig 3. Adjusted Mean SARI stigma score post-intervention for intervention and control groups.**

**Disclosure concerns.** Participants in the intervention group had a significantly lower disclosure concerns score post-intervention compared to baseline, while those in the control group had a significantly higher total score for this domain post-intervention compared to baseline. The mean differences were 6.1 (95% CI 5.6-6.3) for the intervention group and 4.5 (95% CI 5.1 to 4.0) for the control group (Table 2). At baseline, the adjusted mean difference in disclosure concerns scores between the intervention group and control group was 3.45 (95%CI 2.77 to 4.13, p<0.000) (Fig 2, S1 Table 1). Post-intervention, the adjusted mean score for disclosure concerns was lower among respondents in the intervention group compared to the control group, with a mean difference of 7.49 (95%CI 7.93 to 7.05, *p*<0.000) (Fig 3, S2 Table).

**Internalised stigma.** Within-group analysis showed a significant decrease in internalized stigma levels from baseline in the intervention group, while a significant increase was observed in the control group post-intervention. Mean differences of 7.6 (95% CI 7.0-8.2) vs 3.3 (95% CI 3.9-2.6) (Table 2). However, after adjusting for other factors, a mean difference of 1.61 was observed in the control group from baseline to post-intervention (S1 and S2 Tables). Furthermore, at baseline, the adjusted mean internal stigma score for the intervention group was 2.71 points higher in the control group

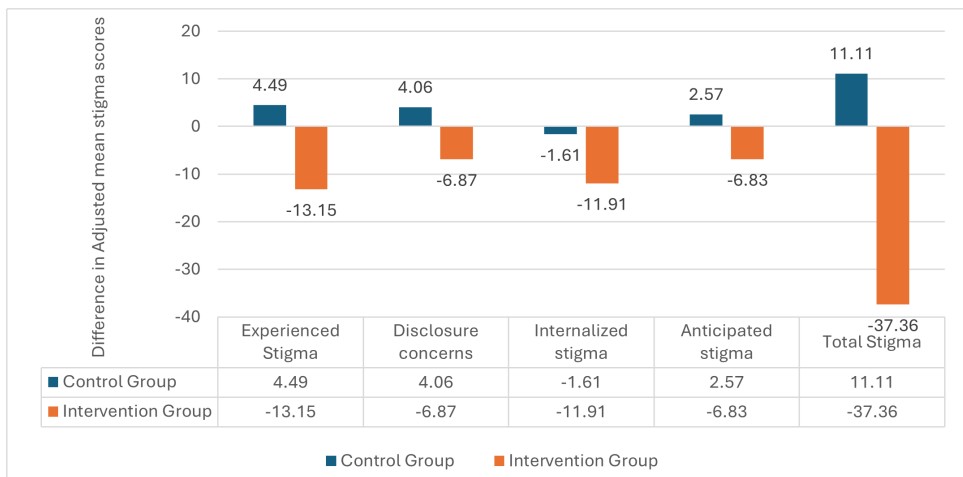

**Fig 4. Within-group differences between baseline and post-intervention in SARI stigma adjusted mean scores.**

**Table 2. Within-group analyses of the impact of the intervention on the intervention group and control group for total scores.**

| | Study group | | | | Control group | | | |
|---|---|---|---|---|---|---|---|---|
| | Baseline (n = 362) | Post-intervention (n = 286) | Difference* | P value | Baseline (n = 273) | Post-intervention (n = 257) | Difference* | P value |
| **Total stigma score** | | | | | | | | |
| Mean | 34.3 | 4.4 | 29.9 | <0.001 | 21.7 | 41.5 | 19.8 | <0.001 |
| 95%CI | 32.4-36.2 | 3.6-5.2 | 27.8-32.0 | | 20.1-23.4 | 40.0-43.1 | 17.5-22.0 | |
| **Experienced stigma** | | | | | | | | |
| Mean | 10.9 | 1.2 | 9.8 | <0.001 | 8.0 | 16.6 | 8.6 | <0.001 |
| 95%CI | 10.1-11.8 | 0.9-1.5 | 8.8-10.6 | | 7.3-8.8 | 15.9-17.4 | 9.7- 7.5 | |
| **Internalized stigma** | | | | | | | | |
| Mean | 8.4 | 0.7 | 7.6 | <0.001 | 4.6 | 7.9 | 3.3 | <0.001 |
| 95%CI | 7.8-9.0 | 9.5-1.0 | 7.0-8.2 | | 4.1-5.1 | 7.4-8.3 | 3.9- 2.6 | |
| **Disclosure concerns** | | | | | | | | |
| Mean | 7.5 | 1.4 | 6.1 | <0.001 | 4.2 | 8.7 | 4.5 | <0.001 |
| 95%CI | 7.0-7.9 | 1.1-1.7 | 5.6-6.3 | | 3.7-4.7 | 8.4-9.0 | 5.1- 4.0 | |
| **Anticipated stigma** | | | | | | | | |
| Mean | 7.5 | 1.1 | 6.4 | <0.001 | 4.9 | 8.3 | 3.4 | <0.001 |
| 95%CI | 7.1-7.9 | 0.8-1.3 | 6.0-6.8 | | 4.4-5.3 | 7.9-8.7 | 4.0- 2.8 | |

* Independent t-test

(95% CI: 1.87 to 3.55, p<0.001) (Fig 2, S1 Table 1). Post-intervention, the intervention group had a significantly lower internal stigma score compared to the control group, with a mean difference of 7.58 (95% CI: 8.10 to 7.07; p<0.001) (Fig 3, S2 Table).

**Anticipated stigma.** Post-intervention, while there was a significant decrease from baseline in the anticipated stigma level in the intervention group, a significant increase in the level of anticipated stigma was observed in the control group, with mean differences of 6.4 (95% CI 6.0-6.8) vs 3.4 (95% CI −4.0-2.8) (Table 2). Comparison between the intervention and control groups showed that at baseline, respondents in the intervention group had a significantly higher adjusted

mean score for anticipated stigma than the control group (2.18 (95% CI 1.57 to 2.80, respectively, p = 0.000) (Fig 2, S1 Table). Conversely, post-intervention, the control group had a significantly higher adjusted mean score than the intervention group, with a mean difference of 7.22 (95% CI 7.65 to 6.78, p < 0.000) (Fig 3, S2 Table).

**SARI scale total score.** The mean total stigma score for respondents in the intervention group was higher at baseline compared to post-intervention, with a mean difference of 29.9 (95%CI 27.8-32.0), p < 0.001 (Table 2). Conversely, the mean total stigma score for respondents in the control group was significantly higher post-intervention compared to the baseline mean with a statistically significant mean difference of 19.8 (95%CI 22.0-17.5) p < 0.001 (Table 2). At baseline, respondents in the intervention group had a significantly higher adjusted mean total score for the SARI Stigma scale than the control group, with an adjusted mean difference of 10.75 (95% CI 8.05 to 13.46, *p* < 0.000) (Fig 2, S1 Table). However, post-intervention, the intervention group had a much lower adjusted mean score than the control group with adjusted mean difference of 37.72 (95% CI 39.43 to 36.01, *p* < 0.000) (Fig 3, S2 Table).

## Multivariate regression analysis of factors influencing the variability in SARI stigma outcome scores

Multivariate regression analyses were conducted to examine the factors influencing the observed differences in stigma scores across the four domains between the intervention and control groups at baseline and post intervention. The results (S1 and S2 Tables) indicate that the determinants varied by domain. At baseline, being employed was significantly associated with lower stigma across all four domains of the SARI scale and the total mean score. Additionally, having primary or secondary education was significantly associated with reduced internalized stigma. Furthermore, participants with Buruli ulcer (BU) reported significantly lower stigma compared to those with leprosy across all the domains and total SARI mean score except anticipated stigma.

Post intervention, two factors- employment and vocational training- consistently impacted two domains as well as the total SARI score. The analysis revealed a statistically significant negative relationship between employment and the total stigma, internal stigma and anticipated stigma scores, indicating that having a paid job is associated with lower levels of perceived stigma. Conversely, vocational training is statistically significantly associated with increase in total stigma and internal stigma scores post-intervention. Additionally, the results show that individuals who have received vocational training tend to have slightly higher concerns about disclosure. While individuals with Buruli ulcer had significantly higher levels of internalised stigma compared to those diagnosed with leprosy, the type of NTD had no significant effect on the total SARI stigma score of participants or other domains. Finally, with the exception of anticipated stigma at baseline and disclosure concern at post intervention, the intercept variance of the total stigma score and the other domains are redundant, meaning that the clusters did not differ significantly from each other in their mean outcome scores. This indicates that any observed differences in these outcomes are not due to variability between clusters but rather are influenced by individual-level factors or the intervention itself.

## Discussion

This study aimed to assess the effect of combined self-help groups on stigma among individuals affected by leprosy or Buruli ulcer. To our knowledge, this is the first randomized controlled trial to evaluate self-help groups as a stigma reduction intervention in BU management, and the second in leprosy management, with the first conducted in Indonesia [23]. The results of this quantitative study demonstrate that participation in self-help groups (SHGs) had a very strong positive effect on stigma, as assessed through the four domains of the SARI stigma scale. Some of the experiences shared by participants and SHG leaders during their routine group meetings provided valuable context to interpret the findings, especially around perceived stigma and social support. While these accounts were not systematically collected as qualitative data, they offer supportive perspectives on the impact of the SHG intervention and are used here for illustrative purposes only.

The baseline differences in some socio-demographic variables especially employment status and educational attainment between the intervention and control groups are noteworthy and may have influenced participants' baseline stigma

levels. A higher proportion of participants in the intervention group were unemployed and had lower levels of formal education compared to the control group.

These socio-economic characteristics are important contextual factors, as they are known to shape individuals' access to resources, health-seeking behavior, and vulnerability to stigma. Consistent with this, our multivariate regression analysis showed that being employed was significantly associated with lower stigma scores across all domains, highlighting the protective role of economic engagement against stigma. In contrast, educational attainment was significantly associated with lower internalized stigma only, suggesting that while education may improve self-perception and coping, it may not shield individuals from enacted or anticipated stigma in the community.

Despite the intervention group's higher baseline stigma and socio-economic disadvantage, they experienced a significant reduction in stigma post-intervention. This underscores the potential of the intervention to improve outcomes even among more vulnerable populations. The increase in stigma scores across all the domains post-intervention in the control groups further strengthens the evidence of the positive impact of SHG participation on stigma. This suggests that, in the absence of supportive interventions like the SHG model, stigma may not remain stable and may in fact worsen. Individuals may experience worsening isolation, internalize more deeply the negative experiences, even in the absence of new events, thereby reinforcing negative self-perceptions and perceived stigma. These findings further highlight the need for targeted interventions to prevent the deterioration of mental and social well-being among affected individuals. This result corroborates those of other non-experimental studies on the positive impact of SHG on stigma among leprosy patients [18,24].

Although the results showed a reduction in the internalized stigma score for the control group after adjusting for other factors post-intervention, the reduction was quite small compared to the intervention group. This finding is consistent with that by Dadun et al. [23], where a reduction in internal and anticipated stigma scores was also observed in the control group post-intervention. While the modest reduction in internalized stigma observed in the control group may be partly attributed to ongoing leprosy awareness campaigns that span both intervention and control areas as part of the national and partners' responses to leprosy control in Nigeria, the overall increase in stigma, particularly in the domains of anticipated and enacted stigma, highlights the limitations of such campaigns when implemented in isolation. Awareness efforts, though valuable for improving knowledge, may not sufficiently address the emotional and behavioral components of stigma, especially in the absence of structured psychosocial support mechanisms.

The increase in stigma within the control group suggests that without targeted interventions like SHGs, stigma can persist or even worsen over time. The findings underscore the need for more participatory, community-centered interventions to effectively mitigate stigma in affected populations [25]

Our findings underscore the impact of the intervention, attributable to the attention and awareness generated within the intervention group through SHG activities. By actively participating in these groups, members received accurate information about the causes and transmission mechanisms of the two diseases, enabling them to dispel harmful myths and misconceptions. Based on information gathered during their regular group meetings and from their interactions with the lay counselors, the nurturing environment of regular meetings, experience sharing, and mutual support promoted social integration, created a sense of community, reduced feelings of isolation, and fostered psychological resilience. Consequently, many felt emboldened to actively participate in social activities within their communities and share their experiences with others outside the group. Similar results have been demonstrated in HIV and mental health programs where support groups help by sharing experiences, encouraging disclosure, reducing stigma and discrimination, boosting self-esteem, enhancing coping skills and psychosocial functioning, and supporting medication adherence and better retention in care [14,26,27]

Our result supports the Social Identity Model of Identity Change (SIMIC), which suggests that well-being and adjustment during life transitions and identity changes are improved when individuals maintain important pre-existing social group memberships or form new ones [28]. SIMIC posits that access to multiple social groups offering various identity

resources can act as a social cure, indicating that strong identification with self-help groups indirectly reduces stigma and directly enhances well-being by providing a platform for accessing these resources [28]. These resources can include social support, civic education, and health literacy, which help redefine participants' self-concept and social image. According to Brown et al [29], important elements of effective stigma reduction interventions include education, personal information, direct challenges to myths, and promotion of empathy through simulations and opportunities for discussion. Our intervention provided the platform to access most of these resources.

We found that being employed is associated with lower total, internal stigma and anticipated mean stigma scores. This finding is consistent with the result of a study in Indonesia on the impact of socio-economic intervention on leprosy-related stigma [10]. Being employed can positively impact stigma by providing economic independence, which enhances self-esteem and self-worth. Employment also fosters social integration through regular interactions with colleagues and the community, reducing feelings of isolation. It offers opportunities for skill development and professional achievement, boosting confidence and countering internalized stigma. Additionally, employment gives individuals a positive social role and identity, helping to overshadow any stigmatized identity. Furthermore, good mental health benefits of employment, such as routine, purpose, and a sense of accomplishment, can further reduce the impact of stigma and improve overall well-being [30–32]. However, it is noteworthy that the relationship between employment and stigma may be bi-directional, as stigma and discrimination can also negatively affect an individual's employability, limiting their opportunities for social and economic participation [33,34]

Contrary to the findings of other studies [23,35] of the positive impact of socio-economic development on stigma reduction, our findings showed negative association between vocational training and total stigma, internalised stigma and disclosure concerns scores. While socio-economic development, such as vocational training, is typically associated with reduced stigma, likely due to its role in empowering individuals, boosting self-esteem, and altering societal perceptions, our results diverge from this trend. Although the association we observed is statistically significant, the effect size is marginal and may not be of practical significance. In addition, this unexpected finding could in part be the influence of a statistical phenomenon rather than the true effect of the intervention. Specifically, the intervention group (many of whom participated in the vocational training), which had significantly higher baseline stigma scores, may have experienced regression to the mean, a tendency for individuals with extreme values to naturally move closer to the average overtime. Nonetheless, further research is needed to validate this finding.

This is the first study to evaluate the impact of stigma reduction intervention involving BU patients. Our results showed a significant reduction in the overall stigma score for participating BU patients including children after implementing the SHG intervention, demonstrating that the intervention is effective in reducing BU-associated stigma. However, BU patients were observed to have higher internalised stigma scores compared to persons affected by leprosy in this study. While we do not claim to know the exact reasons for this, possible explanations include severe skin lesions and deformities associated with BU disease, which may lead to greater social stigma and contribute to higher internalised stigma. Additionally, since leprosy patients make up the majority of the combined self-help groups, the unique challenges faced by BU patients may not be adequately addressed within the groups.

We recommend conducting more randomised controlled stigma reduction intervention studies that will include BU patients to expand the body of evidence. Additionally, comparative studies are needed to understand the unique aspects of BU-related stigma compared to other conditions like leprosy. Qualitative research, such as in-depth interviews and focus groups, can provide personal insights into the experiences of BU patients, informing the development of more effective interventions.

## Limitations of this study

The result of the within-group analysis, although statistically significant, should be interpreted with caution due to the study's limitation. Loss of some unique ID numbers after baseline data collection, coupled with some respondents who

either relocated, died during the period of the intervention, precluded a paired analysis of baseline and post-intervention data of participants. We therefore used an independent sample student t-test rather than paired t-test for within group analysis. The clusters included in this study were purposively selected based on the number of notified cases of leprosy and Buruli ulcer between 2014 and 2018, which could affect the generalization of study findings.

However, we believe this issue would not have altered our overall conclusions. The inclusion of a control group significantly enhanced the robustness and credibility of our findings, representing a notable strength of our study. Moreover, despite a 14% attrition rate, our sample size remained adequate to detect meaningful changes due to the intervention, as it was originally calculated to accommodate an expected 30% attrition rate. This maintained the study's statistical power and ensured that our conclusions about the intervention's effectiveness are reliable and valid. Additionally, the results remained statistically significant even after controlling confounders through multivariate analysis.

The study did not assess or compare visible disability or lesions among participants, which are known contributors to stigma. This omission limits our ability to fully interpret baseline stigma differences between intervention and control groups. Future studies should consider including this clinical parameter to enrich understanding of stigma dynamics. However, this does not undermine the effectiveness of the intervention as the intervention group despite having higher stigma scores at baseline experienced a significant reduction in stigma post intervention, even if they potentially had higher levels of visible deformity.

## Conclusion

We evaluated the effectiveness of combined self-help groups (SHGs) in reducing stigma among persons affected by leprosy or BU. By employing a randomized trial design, our findings provide robust evidence that participation in SHGs leads to significant improvements in stigma-related outcomes compared to standard care alone. This study underscores the potential of SHGs to serve as a scalable and sustainable intervention for stigma reduction. Additionally, expanding this model to other NTDs and diverse geographical settings could further validate the effectiveness and adaptability of SHGs.

## Supporting information

**S1 Checklist. Consort checklist 2010.**
(PDF)

**S1 Data. Inclusivity in global research.**
(DOCX)

**S1 Fig. Consort diagram.**
(TIF)

**S1 Table. Factors influencing the variability of scores at baseline.**
(DOCX)

**S2 Table. Factors influencing variability of scores post-intervention.**
(DOCX)

## Acknowledgments

We gratefully acknowledge the invaluable support provided by the National Tuberculosis, Leprosy, and Buruli Ulcer Control Program, the State Tuberculosis, Leprosy, and Buruli Ulcer Control Program Managers, and the Local Government Tuberculosis, Leprosy, and Buruli Ulcer Control Program Supervisors. Their dedication and assistance were instrumental in the success of this study. We also extend our heartfelt gratitude to all the individuals affected by NTDs who participated

in this research. Their courage and contributions have significantly advanced our understanding and efforts to reduce stigma and improve the quality of life for people impacted by these diseases.

## Author contributions

**Conceptualization:** Chinwe Chika Eze, Ngozi Ekeke, Amaka Onyima-Esmai, Edmund Ndudi Ossai, Joseph Chukwu.

**Data curation:** Chinwe Chika Eze, Edmund Ndudi Ossai.

**Formal analysis:** Chinwe Chika Eze, Wim van Brakel, Edmund Ndudi Ossai.

**Investigation:** Chinwe Chika Eze, Ngozi Ekeke, Precious Henry, Amaka Onyima-Esmai, Edmund Ndudi Ossai, Charles C. Nwafor, Ngozi Murphy-Okpala, Anthony Meka, Okechukwu Ezeakile, Martin Njoku, Francis S. Iyama.

**Methodology:** Chinwe Chika Eze, Ngozi Ekeke, Amaka Onyima-Esmai, Edmund Ndudi Ossai, Charles C. Nwafor, Ngozi Murphy-Okpala, Anthony Meka, Joseph Chukwu.

**Project administration:** Ngozi Ekeke, Precious Henry.

**Supervision:** Chinwe Chika Eze, Precious Henry, Amaka Onyima-Esmai, Chukwuma Anyaike, Daniel Egbule, Joseph Chukwu.

**Validation:** Ngozi Ekeke, Wim van Brakel, Edmund Ndudi Ossai.

**Visualization:** Chinwe Chika Eze, Wim van Brakel.

**Writing – original draft:** Chinwe Chika Eze, Ngozi Ekeke.

**Writing – review & editing:** Chinwe Chika Eze, Ngozi Ekeke, Wim van Brakel, Precious Henry, Amaka Onyima-Esmai, Edmund Ndudi Ossai, Charles C. Nwafor, Ngozi Murphy-Okpala, Anthony Meka, Okechukwu Ezeakile, Martin Njoku, Francis S. Iyama, Chukwuma Anyaike, Daniel Egbule, Joseph Chukwu.

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
