## [Decision Letter · Decision Letter 0]

10 Apr 2025

PONE-D-25-07448Evaluating effectiveness of self-help groups in reduction of stigma among neglected tropical diseases  patients in Southern Nigeria: A Cluster Randomised StudyPLOS ONE

Dear Dr. Eze,

Thank you for submitting your manuscript to PLOS ONE. After careful consideration, we feel that it has merit but does not fully meet PLOS ONE’s publication criteria as it currently stands. Therefore, we invite you to submit a revised version of the manuscript that addresses the points raised during the review process.

confirm the exact number of states involved in the study: six or five. Single digits are better written in words to avoid typos.

We look forward to receiving your revised manuscript.

Kind regards,

Daniel Chukwuemeka Ogbuabor, Ph.D., M.D.

Academic Editor

PLOS ONE

Journal Requirements:

In the figure caption of the copyrighted figure, please include the following text: “Reprinted from [ref] under a CC BY license, with permission from [name of publisher], original copyright [original copyright year].

Additional Editor Comments:

Line 117 - The number of states is five, not six. Single digits are better written in words to avoid typos.

Reviewers' comments:

Reviewer's Responses to Questions

**Comments to the Author**

1. Is the manuscript technically sound, and do the data support the conclusions?

Reviewer #1: Yes

Reviewer #2: Yes

2. Has the statistical analysis been performed appropriately and rigorously? 

Reviewer #1: Yes

Reviewer #2: Yes

3. Have the authors made all data underlying the findings in their manuscript fully available?

Reviewer #1: Yes

Reviewer #2: Yes

4. Is the manuscript presented in an intelligible fashion and written in standard English?

Reviewer #1: Yes

Reviewer #2: Yes

5. Review Comments to the Author

Reviewer #1: This is a commendable study and I congratulate the authors to their work. The results regarding the effectivity of the SHG intervention are truly impressive and I am aware of how much effort must have gone into organising and documenting the SHG in a study population of this size.

The Introduction provides relevant background information about the stigmatisation of NTDs, especially leprosy.

The Methods section clearly explains the study design, and the information about statistical analysis is particularly detailed. However, it is unclear why children with BU are included but children with leprosy are excluded.

Moreover, the qualitative aspects of the study (feedback from participants and SHG leaders) could be included both in the Methods and the Results section. They are brought up in the discussion and provide important insights into the participants' experience of stigma and of the self help groups, so it looks like this is a mixed methods study rather than merely an analysis of the SARI scale. Please see also my more detailed comments in relation to specific parts of the manuscript (below).

In the Data Collection section, I do not think that the other questionnaires need to be mentioned, as they are not presented in the manuscript. Instead, I would like to see more explanation about the SARI scale, including what type of questions were asked, what is the maximum score, and what the different scores mean (low/intermediate/high stigma?).

In the Results, the number of included children should be stated in the table on socio-demographics. I would also like to see information about visible disability, as this is presented as a major driver of stigma and can act as a confounder. It would also be important to see if there are differences in visible lesions/disability between the intervention and control group (which might explain the different levels of stigma at baseline).

Presenting the relative reduction of stigma in the intervention group would highlight the effectiveness of the SHGs.

The Discussion can be developed further by paying more attention to questions arising from the results: How do you explain the difference in stigma scores between the intervention and control group at baseline? And within the control group between pre- and post-intervention? Are these true differences or could they be an indication that the scale is not very reliable?

Line-by-line comments:

Line 72: consider cutting out “the infectious nature of NTDs” as not all NTDs are infectious and fear of contagion is already mentioned in the sentence. If you want to keep it because it relates to shame about having acquired the infection, then please change to “the infectious nature of most NTDs”

Lines 182-86: Why did you expect a prevalence of 50% and not 58% as in the Nigerian study? And why a reduction of 15% and not (over) 20% as in the Indian study?

Lines 184/85: For context, please state that the study was from India

Lines 202ff: Why did you include children with BU but not with leprosy?

Table 1: a) What exactly does “Diagnosis” mean? That they are affected by leprosy/BU or that they are currently under treatment? This is relevant to understand the drop in BU cases from pre- to post-intervention.

b) How many children were included in intervention and control group?

c) A comparison of the prevalence of visible disability in the intervention and the control group would be useful. I understand that this is not socio-demographic data but important nevertheless, e.g. to understand the lower stigma rates in the control group pre-intervention, or why having a job is correlated with less stigma experience (with visible disability as confounder).

Lines 275ff: Can you summarise the (very impressive) impact of the intervention in a couple of sentences at the beginning of this section?

Line 330: …in the control group, with mean differences…

Line 385: I think you can highlight here that your intervention had a very strong positive effect. It’s not only that the results were highly significant, but that stigma was reduced from 42% to 4.6% which is a truly impressive result showing that SHG are highly effective.

Line 394: if you believe that the small reduction in internalized stigma in the control group is due to the ongoing leprosy awareness campaigns, how do you explain the overall increase in stigma in the control group?

Line 398ff: This paragraph is very relevant and interesting as it discusses what makes the SHG so successful. However, the qualitative data this discussion is based on has not been presented in the results section; e.g. that “many felt emboldened to actively participate in social activities”. Including the feedback from the SHG leaders as a study result would make the paper stronger. Alternatively, a part of this paragraph could be moved to the Methods (explaining the objectives of the SHG in more detail) and the discussion here could be based more on the presented literature on HIV and mental health SHG.

Line 424: Please clarify the statement that “being employed is associated with a reduction in total stigma”. Do you mean a reduction between pre- and post-intervention? Or do you mean that being employed is correlated with lower total stigma? Given that discrimination leads to not being employed, the bidirectional relationship between employment and stigma should at least be noted. In addition, visual disability is a potential confounder.

Lines 451ff: If you want to discuss stigma reduction in children, you need to present the data in the results section. Children are not presented as a separate group and therefore there is no evidence of how your study offered insights into their challenges the effectiveness of the intervention in children.

Lines 453-456: as above, these observations (qualitative data) should be mentioned in the results section, if you use them in your discussion.

Reviewer #2: General comments

The paper addresses an important topic which requires public health responses. The findings revealed could create an evidence base for policy and practice.

Specific comments

The paper could be improved if the authors address the following concerns

- The title needs some rephrasing as "...stigma in patients with neglected tropical diseases..."

- Abstract-methodology/principal findings section should clear elaborate the dimensions of stigma targeted by the intervention as well as the total number of samples in the intervention and control arms.

- In lines 76-79, the authors indicated sefl-stigma to be an important dimension of stigma with multiple negative consequences without justifying why their study should focus on this dimension of stigma for intervention in subsequent lines.

- In methods section, there must be a separate section devoted for describing intervention outcomes. What are the intervention outcomes and how are they measured? Why is SARI tool selected among several other alternatives? What are the strengths, weaknesses and statistical attributes of the SARI tool? How is its validity and reliability determined in this study?

- It is also not clear for whom the SARI tool administered during evaluation. Were they members of the SHGs who participated in 24 months peer-support meetings? or Were they general category of leprosy and BU patients recruited from the sampled LGAs? If general patients, how were selected for pre-post evaluation? What was the relationship between SHGs and general category of patients during intervention implementation?

- The relevance of p-values in the table on line 272 is not clear. The table does not also make sense statistically.

6. PLOS authors have the option to publish the peer review history of their article (what does this mean? ). If published, this will include your full peer review and any attached files.

**Do you want your identity to be public for this peer review?** For information about this choice, including consent withdrawal, please see our Privacy Policy .

Reviewer #1: No

Reviewer #2: No

---

## [Author Response · Author response to Decision Letter 1]

7 May 2025

Response have been uploaded as a separate file.

---

## [Editor Report · Decision Letter 1]

20 May 2025

PONE-D-25-07448R1Evaluating effectiveness of self-help groups in reduction of stigma in patients with neglected tropical diseases in Southern Nigeria: A Cluster Randomised StudyPLOS ONE

Dear Chinwe Chika Eze,

Thank you for submitting your manuscript to PLOS ONE. After careful consideration, we feel that it has merit but does not fully meet PLOS ONE’s publication criteria as it currently stands. Therefore, we invite you to submit a revised version of the manuscript that addresses the points raised during the review process.

**ACADEMIC EDITOR: **

Line 208: Do the authors have baseline data on the eleven children with leprosy? It seems the authors did not collect baseline data on few children with leprosy (n=11) but included them in the implementation phase for ethical concerns. If this is the case, the authors should explain this gap in the manuscript and acknowledge it as a limitation. Regarding the authors’ explanation in their response to reviewers’ comments, the reason, “Owing to the epidemiology of leprosy in Nigeria, most persons affected are usually adults,” is insufficient to exclude children with leprosy from participating in the study. Lines 187-191: The authors should explain the contextual realities in the study setting justifying the conservative estimate of 50% and 15% instead of the prevalence of 58% in Nigeria and at least 20% in India. In the absence of strong justification, the study should adopt the prevalence from the empirical evidence. The World Bank study, reference 21, is irrelevant to the sample size determination in this study and needs to be expunged.

We look forward to receiving your revised manuscript.

Kind regards,

Daniel Chukwuemeka Ogbuabor, Ph.D., M.D.

Academic Editor

PLOS ONE

Journal Requirements:

Additional Editor Comments (if provided):

Line 208: Do the authors have baseline data on the eleven children with leprosy? It seems the authors did not collect baseline data on few children with leprosy (n=11) but included them in the implementation phase for ethical concerns. If this is the case, the authors should explain this gap in the manuscript and acknowledge it as a limitation. Regarding the authors’ explanation in their response to reviewers’ comments, the reason, “Owing to the epidemiology of leprosy in Nigeria, most persons affected are usually adults,” is insufficient to exclude children with leprosy from participating in the study.

Lines 187-191: The authors should explain the contextual realities in the study setting justifying the conservative estimates of 50% and 15% instead of the prevalence of 58% in Nigeria and at least 20% in India. In the absence of strong justification, the study should adopt the prevalence from the empirical evidence. The World Bank study, reference 21, is irrelevant to the sample size determination in this study and needs to be expunged.

---

## [Author Response · Author response to Decision Letter 2]

29 May 2025

Reviewer comment

Line 208: Do the authors have baseline data on the eleven children with leprosy? It seems the authors did not collect baseline data on few children with leprosy (n=11) but included them in the implementation phase for ethical concerns. If this is the case, the authors should explain this gap in the manuscript and acknowledge it as a limitation. Regarding the authors’ explanation in their response to reviewers’ comments, the reason, “Owing to the epidemiology of leprosy in Nigeria, most persons affected are usually adults,” is insufficient to exclude children with leprosy from participating in the study.

Author response

Thank you for your thoughtful observation. We would like to clarify that baseline and post-intervention data were indeed collected for all participants, including the eleven children affected by leprosy across the intervention and control groups.

As noted in the paper, this intervention was part of a larger randomised controlled study. While our original inclusion criteria for the study specified that children affected by leprosy would not be part of the study population, these children were identified during the community sensitization and mobilization activities. Given their status as persons affected by leprosy and in line with ethical principles of equity and beneficence, we felt it was important not to exclude them from benefiting from the intervention, particularly the self-help group component targeting stigma reduction-especially, since their counterparts affected by Buruli ulcer were part of the study.

Recognizing that this introduces a minor inconsistency between the inclusion criteria and implementation, we acknowledged this exception and its implication under the study limitation section of the manuscript.

We hope this clarification addresses your concern and we appreciate your commitment to ensuring the rigor and transparency of the study.

Reviewer comment

Lines 187-191: The authors should explain the contextual realities in the study setting justifying the conservative estimate of 50% and 15% instead of the prevalence of 58% in Nigeria and at least 20% in India. In the absence of strong justification, the study should adopt the prevalence from the empirical evidence. The World Bank study, reference 21, is irrelevant to the sample size determination in this study and needs to be expunged.

Author response

Thank you once again for the valuable feedback. As previously noted, we were conservative while conceptualizing the study as we do not want to be ambitious by committing to the same prevalence and effect size especially where the contexts differ even within Nigeria. However, in line with your suggestion, we have updated our assumptions accordingly and confirm that our sample size remains adequate. In addition, we agree that the World Bank study added no value to the sample size determination and have removed it as advised.

---

## [Editor Report · Decision Letter 2]

17 Jun 2025

PONE-D-25-07448R2Evaluating effectiveness of self-help groups in reduction of stigma in patients with neglected tropical diseases in Southern Nigeria: A Cluster Randomised StudyPLOS ONE

Dear Dr. Eze,

Thank you for submitting your manuscript to PLOS ONE. After careful consideration, we feel that it has merit but does not fully meet PLOS ONE’s publication criteria as it currently stands. Therefore, we invite you to submit a revised version of the manuscript that addresses the points raised during the review process.

We look forward to receiving your revised manuscript.

Kind regards,

Daniel Chukwuemeka Ogbuabor, Ph.D., M.D.

Academic Editor

PLOS ONE

Journal Requirements:

**Additional Editor Comments:**

Generally, if data from a population outside the inclusion criteria for a study is collected, it should not be used unless there are specific ethical and methodological justifications. In the current study, it seems ethically inappropriate to have excluded children with leprosy from the study simply because leprosy was commoner among adults. The authors report that they identified 11 children with leprosy who met the inclusion criteria during community mobilisation, which is a sufficient reason to revise the study's inclusion criteria for people with leprosy. The authors further report that they collected baseline and post-intervention data from these 11 children with leprosy, and "their baseline and post-intervention data were collected and analysed in the same way as for other participants in the respective study groups." Excluding children with leprosy from the overall analysis after collecting pre- and post-intervention data from them is unethical and unacceptable. First, the study included children with Buruli ulcers. Secondly, the authors have an opportunity to revise the inclusion criteria. Thirdly, without reporting the findings on children with leprosy, we cannot ascertain whether excluding children with leprosy had any impact on the overall analysis or validity of the study. Therefore, the authors may revise the inclusion criteria and include the children with leprosy in the data analysis, results, and discussion.

---

## [Author Response · Author response to Decision Letter 3]

17 Jun 2025

Thank you very much for the valuable feedback.

As recommended, we have revised the inclusion criteria to incorporate children with leprosy. This adjustment is appropriate, as the study already had ethical approval for the inclusion of children with Buruli ulcer, and the contextual realities are such that children affected by leprosy are naturally integrated into self-help groups.

Excluding them would have been ethically inappropriate, as rightly noted. The data collected from the 11 children with leprosy were included in the analysis and results—both baseline and post-intervention—as presented in the manuscript.

However, their data were not highlighted separately in the discussion section, as the analysis and interpretation were conducted at the group level across all age categories, without a specific focus on children.

---

## [Editor Report · Decision Letter 3]

20 Jun 2025

Evaluating effectiveness of self-help groups in reduction of stigma in patients with neglected tropical diseases in Southern Nigeria: A Cluster Randomised Study

PONE-D-25-07448R3

Dear Chinwe Chika Eze,

We’re pleased to inform you that your manuscript has been judged scientifically suitable for publication and will be formally accepted for publication once it meets all outstanding technical requirements.

Kind regards,

Daniel Chukwuemeka Ogbuabor, Ph.D., M.D.

Academic Editor

PLOS ONE
---

## [Editor Report · Acceptance letter]

PONE-D-25-07448R3

PLOS ONE

Dear Dr. Eze,

I'm pleased to inform you that your manuscript has been deemed suitable for publication in PLOS ONE. Congratulations! Your manuscript is now being handed over to our production team.

Kind regards,

on behalf of

Dr. Daniel Chukwuemeka Ogbuabor

Academic Editor

PLOS ONE